# Intermittency in fluid and MHD turbulence analyzed through the prism of moment scaling predictions of multifractal models

Annick Pouquet[1], Raffaele Marino[2], Hélène Politano[3], Yannick Ponty[4], and Duane Rosenberg[5]

[1]National Center for Atmospheric Research, P.O. Box 3000, Boulder, CO 80307, USA

[2]Laboratoire de Mécanique des Fluides et d'Acoustique, CNRS, École Centrale de Lyon, Université Claude Bernard Lyon 1, INSA de Lyon, F-69134 Écully, France

[3]Université Côte d'Azur, CNRS, LJAD, France

[4]Université Côte d'Azur, CNRS, Observatoire de la Côte d'Azur, Laboratoire Lagrange, France

[5]Cooperative Institute for Research in the Atmosphere (CIRA), NOAA/OAR Global Systems Laboratory, Colorado State University, 325 Broadway Boulder, Fort Collins, CO 80305, USA

**Correspondence:** Annick Pouquet (pouquet@ucar.edu)

**Abstract.** In the presence of waves due *e.g.* to gravity, rotation or a quasi-uniform magnetic field, energy transfer time-scales, spectra and physical structures within turbulent flows differ from the fully developed fluid case, but some features remain such as intermittency or quasi-parabolic behaviors of normalized moments of relevant fields, for the most part in that intermediate regime where waves and nonlinear eddies strongly interact. After reviewing some of the roles intermittency can play in various geophysical flows, we present results of direct numerical simulations at moderate resolution and run for long times. We show that the power-law scaling relations between kurtosis $K$ and skewness $S$ found in multiple and diverse environments can be recovered using a selection of existing multifractal intermittency frameworks. Indeed, in the specific context of the She-Lévêque model (1994) generalized to MHD and developed as a two-parameter system in Politano and Pouquet (1995), we find that a parabolic $K(S)$ law can be recovered for maximal intermittency involving the most extreme dissipative structures.

## 1 Introduction

A word-frequency study performed on research papers centered on a variety of atmospheric issues indicated that the most frequent cloud-controlling factor is turbulence (Siebesma et al. (2009); see also Pumir and Wilkinson (2016)), likely because of its ubiquity but also because it could presumably explain a multitude of somewhat puzzling phenomena that occur at small scale, be it only that of order-unity dissipation at high Reynolds number. More recently, fully developed turbulence (FDT) has been associated with the barotropic state of large-scale atmospheric turbulence, with the multiplicative effect due to turbulence in the occurence of the acceleration of the jet stream, and the rapid intensification of hurricanes (Shepherd, 2020; Emanuel et al., 2023; Shaw and Miyawaki, 2024). A similar study for plasma physics might reveal the same feature, namely that the complexity of nonlinear phenomena is the dominant property impeding the development of wide-encompassing theoretical and modeling techniques of small-scale behavior, thus making the much-needed prediction of disruptions in fusion plasmas difficult, even though it is essential.

Observations of magnetohydrodynamic (MHD) and plasma turbulence in space physics are numerous, with consistent progress in the resolution of satellite instrumentation and with now the exploration of the kinetic regime (Fox et al., 2016; Muller, D. et al., 2020). The access to small-scale dynamics through newly-launched spacecrafts allows for example for a direct evaluation of the dissipation rate spectrum through the measurement of the current and electric field using the Magnetospheric MultiScale Mission (MMS, He et al. (2019)). It has been known for a long time that vortex sheets observed in the first direct numerical simulations (DNS) of turbulence using pseudo-spectral methods could roll-up into vortex filaments (Patterson and Orszag, 1971; Siggia and Patterson, 1978) (see Douady et al. (1991)) for experimental evidence), whereas in MHD the dynamics leads to complex current and vorticity structures stemming from the sheet destabilization observed in DNS in two dimensions (2D) (Matthaeus and Lamkin, 1986; Pouquet et al., 1986), and leading to various reconnection processes and possible singularities (Friedel et al., 1997; Kerr and Brandenburg, 1999; Cartes et al., 2009), a topic however which will not be covered in this review (see also *e.g.* Bhattacharjee (2004); Mininni et al. (2008); Zweibel and Yamada (2009); Daughton et al. (2011); Zhdankin et al. (2013); Lazarian et al. (2020); Oka et al. (2022) for more details). Reconnection and the intermittency associated with singularities have been related (Osman et al., 2014), including at high cross helicity (Smith et al., 2009), and can lead to plasma heating (Marino et al., 2008). Lastly, studies were made to determine the possible development of singular structures in fluids and plasmas in the limit of infinite Reynolds number, but the problem remains open.

Furthermore, new accurate observations of the magnetic field of the Earth have been obtained recently from global ocean circulation measurements, leading potentially to a better understanding of oceanic tides, of ionosphere-magnetosphere interactions and of their variabilities (Hornschild et al., 2022). Thus, one of the marked property of velocity and magnetic fields is that of intermittency (and ensuing anomalous scaling), that is the presence of strong localized structures. These structures can be identified as vortex filaments, as Alfvén vortices which are observed in the solar wind (Wang et al., 2019), or current sheets which undergo instabilities such as Kelvin-Helmholtz (KH) (see Barkley et al. (2015) for a recent review of KH), reconnection and thus dissipation (Matthaeus and Montgomery, 1981; Uzdensky et al., 2010; Faganello and Califano, 2017; Adhikari et al., 2021)). An abundance of observations of our close environment points to a complex suite of systems and structures that include turbulence and nonlinearities in MHD and plasma instabilities, displaying as well anomalous scaling and dissipation (see *e.g.* for recent reviews Matthaeus et al. (2015); Chen (2016); Galtier (2018); Schekochihin (2022); Balasis et al. (2023); Marino and Sorriso-Valvo (2023)). Intermittent dissipation in the MHD range has been shown to lead to beam acceleration in the magnetosphere at ionic scales and below (Sorriso-Valvo et al., 2019)), and particle acceleration has also been observed with MMS in the vicinity of a reconnection X-line, leading also to strong turbulence (Ergun et al., 2020).

There are of course plenty other manifestations of intermittency, *e.g.* through non-Gaussian wings on Probability Distribution Functions (PDFs) for Eulerian and Lagrangian fields. Thus, one way to characterize intermittency in turbulence is through the dual observation of large-scale structures separated by sharp active gradients both for fluids and MHD, particularly noticeable in 2D (Kinney et al., 1995; Meneguzzi et al., 1996; Matthaeus et al., 2015). Another way to quantify the degree of intermittency of a flow is to measure the anomalous exponents of structure functions, *i.e.* measure a departure from self-similarity, as done in the solar wind (Burlaga, 1991) and in DNSs (Politano et al., 1995). MHD intermittency models were built (Grauer et al., 1994; Politano and Pouquet, 1995) (see also Horbury and Balogh (1997)) to explain the observed behavior, but one difficulty resides

in the necessity of having a vast amount of data. In this context, after giving the equations in the next section, we shall analyze in §3 numerical results on the third and fourth-order normalized moments in several systems run at moderate Reynolds numbers for long times, and give a justification of power-law behavior between moments in the framework of turbulence models in §4,

together with, in fact, an extension to scaling laws for arbitrary orders. We mention other frameworks for the study of such intermittency in §5, and conclude in the last section.

## 2   Equations, parameters and numerical set-up

The incompressible equations for rotating stratified flows in the Boussinesq incompressible framework are:

$$\partial_t \mathbf{u} + \mathbf{u} \cdot \nabla \mathbf{u} = -\nabla p - N\theta \hat{z} + 2\mathbf{u} \times f_0 \hat{z} + \nu \nabla^2 \mathbf{u} + \mathbf{F}_u \ , \ \ \partial_t \theta + \mathbf{u} \cdot \nabla \theta = Nw + \kappa_0 \nabla^2 \theta + F_\theta \ , \ \ \nabla \cdot \mathbf{u} = 0 \ , \quad (1)$$

with $\mathbf{u}, \theta$ the velocity and the temperature fluctuations (in velocity units here), $w$ the velocity in the direction of imposed gravity and/or rotation (here, the vertical $z$ direction), $p$ the pressure, $N$ and $f_0/2$ the Brunt-Väisälä and rotation frequencies, and $\nu, \kappa_0$ the viscosity and thermal diffusivity, taken equal (unit Prandtl number). $\mathbf{F}_u, F_\theta$ are forcing terms. For $N = 0, f_0 = 0$, one recovers the Navier-Stokes (NS) equations with a passive scalar. We also write the magnetohydrodynamic (MHD) equations, again for the incompressible case, and with $\mathbf{b}$ the induction in Alfvén velocity units and $P = p + |\mathbf{b}|^2/2$ the total pressure:

$$[\partial_t + \mathbf{u} \cdot \nabla]\mathbf{u} \equiv D_t \mathbf{u} = -\nabla P + \mathbf{b} \cdot \nabla \mathbf{b} + \nu \Delta \mathbf{u} + \mathbf{F}_u \ , \ \ [\partial_t + \mathbf{u} \cdot \nabla]\mathbf{b} \equiv D_t \mathbf{b} = \mathbf{b} \cdot \nabla \mathbf{u} + \eta \Delta \mathbf{b} \ , \ \ \nabla \cdot \mathbf{b} = 0 \ , \ \ P_M = \nu/\eta \ ; \ (2)$$

here, $\eta$ is the magnetic diffusivity, and $P_M = \nu/\eta$ the magnetic Prandtl number. The results described herein have been obtained integrating numerically these equations with pseudo-spectral accuracy using the GHOST (Rosenberg et al., 2020) or CUBBY (Ponty et al., 2005) codes. In the absence of dissipation ($\nu = 0, \ \eta = 0, \ \kappa_0 = 0$), the total energy is conserved as well as cross-helicity and magnetic helicity in MHD, and potential vorticity in the stratified case (see §3.3 for a definition of helicity).

Given a typical large scale taken as the integral scale $L_0$, and a characteristic *r.m.s.* velocity at that scale, $u_0$, one defines the kinetic and magnetic Reynolds numbers and the Froude and Rossy numbers, $Fr, Ro$ in a standard way, namely:

$$R_V = \frac{u_0 L_0}{\nu} \ , \ R_M = \frac{u_0 L_0}{\eta} \ , \ \ Fr = \frac{u_0}{L_0 N} \ , \ Ro = \frac{u_0}{L_0 f_0} \ ; \ R_B = Ro Fr^2 \ , \ R_\lambda = \frac{\lambda}{L_0} R_V \ , \ Ri_g = N(N - \partial_z \theta)/[\partial_z u_\perp]^2 \ . \ (3)$$

$Fr, Ro$ measure the wave period *vs.* the turn-over time $\tau_{NL} = L_0/u_0$, and $R_B$ the intensity of the waves. Are also defined the Taylor Reynolds number $R_\lambda$ based on the Taylor scale $\lambda = \sqrt{\langle u^2 \rangle / \langle \omega^2 \rangle}$, with $\boldsymbol{\omega} = \nabla \times \mathbf{u}$ the vorticity, and the gradient

Richardson number $Ri_g$. The kinetic and magnetic energies are $E_V = \langle \mathbf{u}^2 \rangle/2, E_M = \langle \mathbf{b}^2 \rangle/2$, and $\mathbf{u} \cdot \mathbf{F}_V$ is the kinetic energy input. The point-wise dissipation rates of kinetic and magnetic energy are $\epsilon_v(\mathbf{x}) = \mathbf{u} \cdot \partial_t \mathbf{u} \ , \epsilon_m(\mathbf{x}) = \mathbf{b} \cdot \partial_t \mathbf{b}$. They can be expressed in terms of the symmetric part of the velocity gradient tensor, $S_{ij}$, and of $j^2$, with $\mathbf{j} = \nabla \times \mathbf{b}$ the current density:

$$S_{ij}(\mathbf{x}) = \frac{\partial_j u_i(\mathbf{x}) + \partial_i u_j(\mathbf{x})}{2} \ , \ \epsilon_v(\mathbf{x}) = 2\nu \Sigma_{ij} S_{ij}(\mathbf{x}) S_{ij}(\mathbf{x}) \ , \ \ \epsilon_m(\mathbf{x}) = \eta j^2(\mathbf{x}) \ . \quad (4)$$

Finally, the skewness and excess kurtosis (both zero for a Gaussian) for a scalar field $f$, and the flatness $F_f$ are

$$S_f = \langle f^3 \rangle / \langle f^2 \rangle^{3/2} \ , \ K_f = \langle f^4 \rangle / \langle f^2 \rangle^2 - 3 = F_f - 3 \ , \ K_f(S_f) \sim S_f^{\alpha_f} \ . \quad (5)$$

In the following sections, variations of $\alpha_f$ with parameters will be succinctly analyzed for several turbulence fields and settings.

# 3 Numerical data on $K(S) \sim S^\alpha$ behavior for a few geophysical turbulent flows

## 3.1 The fluid case

Many articles and books have been devoted to an in-depth analysis of turbulence – and perhaps its main distinctive property, that of intermittency– both from a statistical point of view and from a geometrical one (see *e.g.* Kolmogorov (1962); Arnold (1963); Frisch (1995); Chapman and Watkins (2001); Lovejoy and Schertzer (2013); Arnold and Khesin (2021); Benzi and Toschi (2023))[1]. Intermittency is found in the inertial range as well as at the onset of the dissipative range (Kraichnan, 1967a; Sreenivasan, 1985), and it is also present in quantum turbulence (Müller et al., 2021), or MHD turbulence in the laboratory and the cosmos (Zel'dovich et al., 1983). Recall that, in the presence of waves, there are three distinct regimes (see Pouquet et al. (2019) for a recent detailed study in the context of rotating stratified flows), whereby the waves are faster (quasi-linear regime), or the eddies are faster (fully turbulent regime), or the intermediate state where both strongly interact. This leads to variable efficiency of energy transfer and enhanced intermittency, as in the form of large excess kurtosis in a reduced volume of the fluid (Marino et al., 2022). The resulting complexity of turbulent flows has been described using a multitude of tools such as stochastic Langevin equations, self-organized criticality or multifractals, and the presence of anisotropy due to an external agent such as gravity, rotation or a uniform magnetic field has proven useful to take into consideration (see *e.g.* Bak et al. (1987); Sreenivasan and Antonia (1997); Bramwell et al. (2000); Chapman and Watkins (2001); Sagaut and Cambon (2008); Nazarenko (2011); Lovejoy and Schertzer (2013)).

In this context, and associating here intermittency with non-Gaussian behavior through a measure of third and fourth-order normalized moments, $K$ and $S$, we briefly give numerical results showing the ubiquity of $K(S) \sim \kappa S^\alpha$ scaling in turbulent flows, with variable $\alpha$s, and stressing the following examples: Navier-Stokes fluid turbulence, stratified flows without or with rotation as in the atmosphere and oceans, and MHD in the fast dynamo regime.

Perhaps the first instantiation of a $K(S) \sim S^2$ law was derived analytically in Longuet-Higgins (1963) in the context of the ocean, and verified observationally in Ochi and Wang (1984) for coastal waves. It was viewed as a correction to a Gaussian law for small departures from normality, and with $K \geq 0$, a quadratic term in $K(S)$ expansion arises at lower order, together with a constant term. A somewhat surprising result of later studies was that this scaling persisted in some instances for regimes that were strongly turbulent, discovered for geo-fluids as in the troposphere and the boundary layer (Mahrt, 1989; Lenschow et al., 1994, 2012; Lyu et al., 2018), or in the mesosphere (Chau et al., 2021), for ocean and climate dynamics (Sardeshmukh and Sura, 2009; Sardeshmukh and Penland, 2015), as well as for diverse plasmas experiments (Labit et al., 2007; Krommes, 2008). Several studies in a variety of physical contexts ensued, indicating an ubiquity for this law, although a strict parabola was hard to determine (see Pouquet et al. (2023); Ponty et al. (2025) for recent accounts).

The pure fluid case, somewhat surprisingly, was examined only recently to our knowledge (Pouquet et al., 2023) (see also Sattin et al. (2009)). In Sreenivasan and Antonia (1997), one finds a compilation of skewness and flatness up to Taylor Reynolds number in excess of $3 \times 10^4$, for a variety of flows, experimental, numerical and in the atmospheric boundary layer (see their Figures 5 and 6). By digitalizing the data, making log-log fits and selecting points with $R_\lambda \geq 660$, one finds a fit $K \approx S^{2.34}$.

---

[1]For a recent review on turbulence and intermittency in the presence of waves, see *e.g.* Nazarenko (2011).

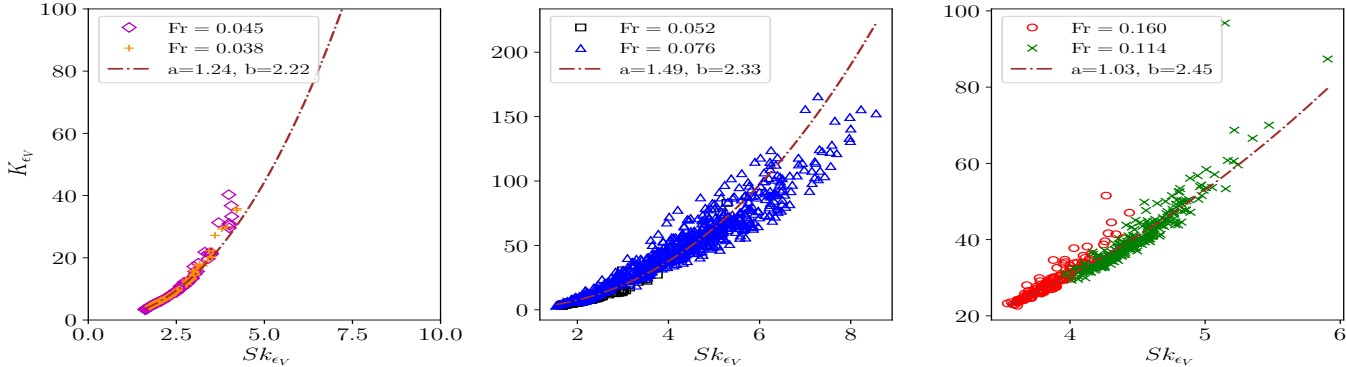

**Figure 1.** Stratified turbulence: Flatness-skewness plots for $\epsilon_v$ for several Froude numbers (see insets with $Fr$ and $(a, b)$ parameters assuming a fit $F \sim a * Sk^b$. Note the different scales on the axes, and in particular the high flatness values for the run with $Fr \approx 0.076$ (middle).

It will be of interest to redo this compilation with more recent experiments, but this already tells us that a pseudo-parabolic scaling $K(S) \sim S^\alpha$, $\alpha \approx 2$ is present for fluid turbulence, as shown as well in numerical simulations of the Navier-Stokes equations with a passive scalar (see Table 1 in Pouquet et al. (2023), a paper denoted hereafter PRM2). Note that the analysis in PRM2 was done rather in terms of the variation with flows or with governing parameters, say the Froude number $Fr$, of the coefficient assuming a parabolic fit, *viz.* $K \sim a(Fr)S^2$, whereas in the present paper we do not assume *a priori* the power-law scaling between $K$ and $S$, and instead search for $\alpha$. We observed in PRM2 that quasi-parabolæemerge, for example for the vertical buoyancy flux $\langle w\theta \rangle$. Also, it was found that $K(S)$ statistics of local square vorticity and local dissipation differ somewhat, in particular at moderate $R_V$ values, but such statistics are shown in Donzis et al. (2008) to be quite similar for the most extreme events, defined as having $10^4$ times the mean dissipation at high $R_\lambda$. It will thus be of interest to extend this type of studies to substantially higher $R_V$.

## 3.2 Stratified flows in the presence or not of rotation

Taking now into account stable stratification, as is found in the atmosphere and the ocean, we plot in Fig. 1 for several Froude numbers (see insets), the power-law fits for the flatness, *viz.* $F(S) \sim S^{\alpha_\epsilon}$ for the kinetic energy dissipation $\epsilon_v$, a good indicator of clear-air turbulence (Storer et al., 2019); this leads to an exponent $\alpha_\epsilon$ that increases continuously with $Fr$, from $\approx 2.22$ to 2.45 (parameters for the runs are given in Table 1 of PRM2). The highest values of both $S_{\epsilon_v}$ and $F_{\epsilon_v}$ are reached for the run with $Fr \approx 0.076$ corresponding to the strongest intermittency of the vertical velocity in particular (Feraco et al., 2018; Marino et al., 2022), strong local dissipation and associated strong localized shear layers.

When combining rotation and stratification of comparable magnitude as found in the ocean ($N/f_0 \approx 5$), we can observe in Fig. 2 (left) for runs with quasi-geostrophic (QG) forcing (see Table 1 of PRM2 for run specifications) a sharp transition in the exponent of the $K(S) \sim S^\alpha$ fits for the buoyancy flux around $R_B \approx 25$, corresponding to an average gradient Richardson number $\langle Ri_g \rangle \approx 1.5$, close to that for a KH transition to instability. As noted before and as opposed to associating the transition

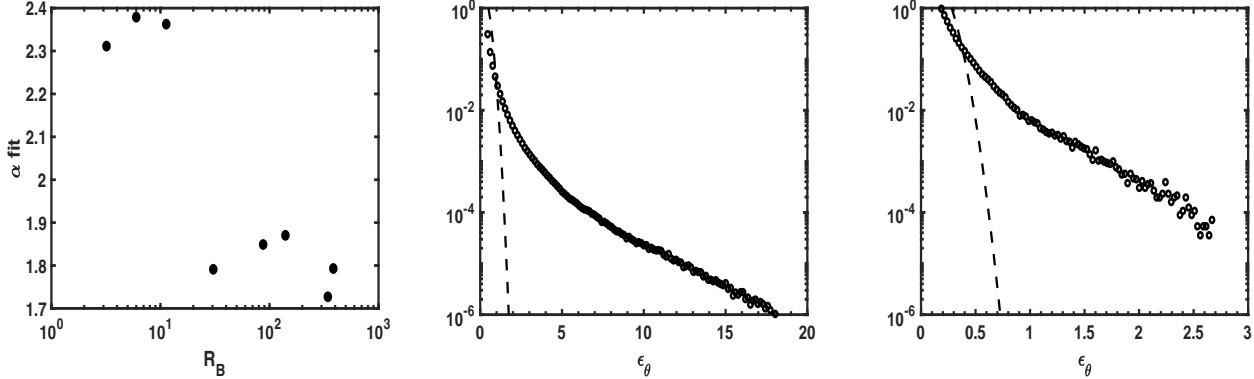

**Figure 2.** Quasi-geostrophic (QG) turbulence: *Left:* Variation with $R_B$ of exponent for $K(S) \sim S^\alpha$ scaling of the kinetic energy dissipation, using a QG forcing. Note the sharp transition which occurs for $R_B \approx 20$ corresponding to $\langle Ri_g \rangle \approx 1$. *Center and right:* PDFs of potential energy dissipation $\epsilon_\theta$ for runs with $R_B = 3$ (run Q1, center) and $R_B = 385$ (run Q8, right; see Table 1 and equation 3 in PRM2 where PDFs for $\epsilon_v$ are given); dashed lines are for equivalent Gaussians, and lin-log coordinates are used, giving plausibility to exponential fits (see text).

to a parabolic law with measurably different coefficients, the transition now is linked to change in the power-law scaling itself, as backed up by the theoretical investigation in §4 in particular in the context of the generalized She-Lévêque models. This also points to the importance of the occurrence of turbulence at small scale once the Ozmidov scale is larger than the Kolmogorov scale, $\ell_{Oz} \geq \eta_K$ with $\ell_{Oz} = \sqrt{\epsilon_V/N^3}$, $\eta_K = [\epsilon_V/\nu^3]^{-1/4}$. We also give in Fig. 2 the PDFs of the potential energy dissipation $\epsilon_\theta = \kappa_0 \langle |\nabla \theta|^2 \rangle$ for the QG runs Q1 (center) and Q8 (right) with buoyancy Reynolds numbers $R_B$ of 3.2 and 385 (see Table 1 in PRM2). The respective fits ($\approx 0.1 \exp^{-2.7\epsilon_\theta}$ and $\approx 0.0008 \exp^{-0.37\epsilon_\theta}$) are in agreement with the expected increase in small-scale structures and dissipation as $R_B$ grows and a fully turbulent regime is reached.

### 3.3 Coupling to a magnetic field in MHD: fast dynamos in the ABC, Roberts and Taylor-Green flows

The dynamo problem is that of the growth of magnetic fields due to either, at small scale, chaotic streamlines of the velocity or, at large scales, the kinetic helicity content of the flow, where $H_V = \langle \mathbf{u} \cdot \boldsymbol{\omega} \rangle$ (Steenbeck et al., 1966; Moffatt, 1969; Zel'dovich et al., 1983; Brandenburg and Subramanian, 2005), and it plays an essential role in the solar context in the presence of convection (see *e.g.* Ponty et al. (2001)). Both the cross-helicity $H_C = \langle \mathbf{u} \cdot \mathbf{b} \rangle$ (Pouquet et al., 1986; Yokoi, 2013), and the magnetic helicity $H_M = \langle \mathbf{A} \cdot \mathbf{b} \rangle$, with $\mathbf{b} = \nabla \times \mathbf{A}$, also play a role, the latter in the nonlinear saturation of the large-scale dynamo associated with an inverse cascade of $H_M$ (Pouquet et al., 1976)[2]. In fact, with sufficient large-scale separation, a dynamo can occur with $H_V \equiv 0$ overall but with sufficient local fluctuations (Gilbert et al., 1988). The dynamo can also be sub-critical because the growing magnetic seed will alter the flow and reduce the turbulence (Ponty et al., 2007; Mannix et al., 2022). The resulting

---

[2]See Enciso and Peralta-Salas (2023) for a recent review on Beltrami fields and the central role of the helicity invariants $H_V, H_M$ and $H_C$. The conservation of $H_M$ was first derived in Woltjer (1958) in the context of force-free fields.

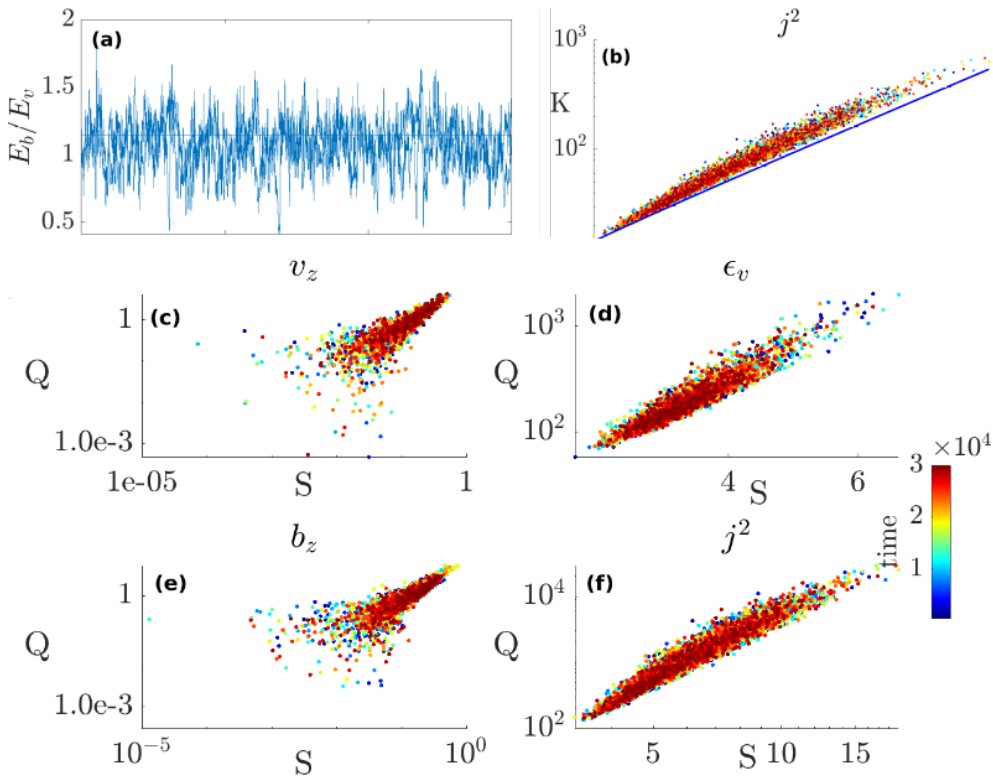

**Figure 3.** Fast dynamos in MHD for run GOR2 with $R_V \approx 445$. (a) $E_M/E_V(t)$, and (b) $K(S)$ for $j^2$. We also give $Q(S)$ for the same run for $j^2$ (b), $v_z$ (c), $\epsilon_v$ (d), $b_z$ (e) and finally (f), for $j^2$ again, this time in log-log.

3D turbulent system is made-up of current and vorticity sheets, rolling-up around the local mean magnetic field and with a strong twist of **b** across the sheet (Mininni et al., 2006; Ponty and Plunian, 2011; Homann et al., 2014); see also Uzdensky et al. (2010); Lazarian et al. (2020); Oka et al. (2022).

(Quasi)-parabolic $K(S)$ laws in MHD have been found both in laboratory plasmas and in the cosmos (Labit et al. (2007); Krommes (2008); Osmane et al. (2015)). Recently, variations of $\alpha$ with parameters have been discussed briefly in the context of the classical She-Lévêque (SL) model as found in the fast dynamo context (Ponty et al., 2025), and we are expanding on these results presently for generalized SL models (see equ. (8) analyzed in §4.1), as well as for higher-order moment ratios.

We now give new results for runs already analyzed in part in Ponty et al. (2025), as well as for two new runs with a so-called
Roberts flow with a helical forcing; these are runs GOR1 and GOR2, with respective Reynolds numbers of 147 and 445 (and $R_\lambda$ of 38 and 66); the runs are performed on grids of $64^2 \times 128$ and $128^2 \times 256$. These flows are well resolved (the dissipation scale is more than twice the numerical cut-off according to the Kaneda criteria), and they are run for long times (15000 and 2000 $\tau_{NL}$ *resp.*). However, we note that the energy spectra (not shown) are not yet sufficiently developed (see also Ponty and

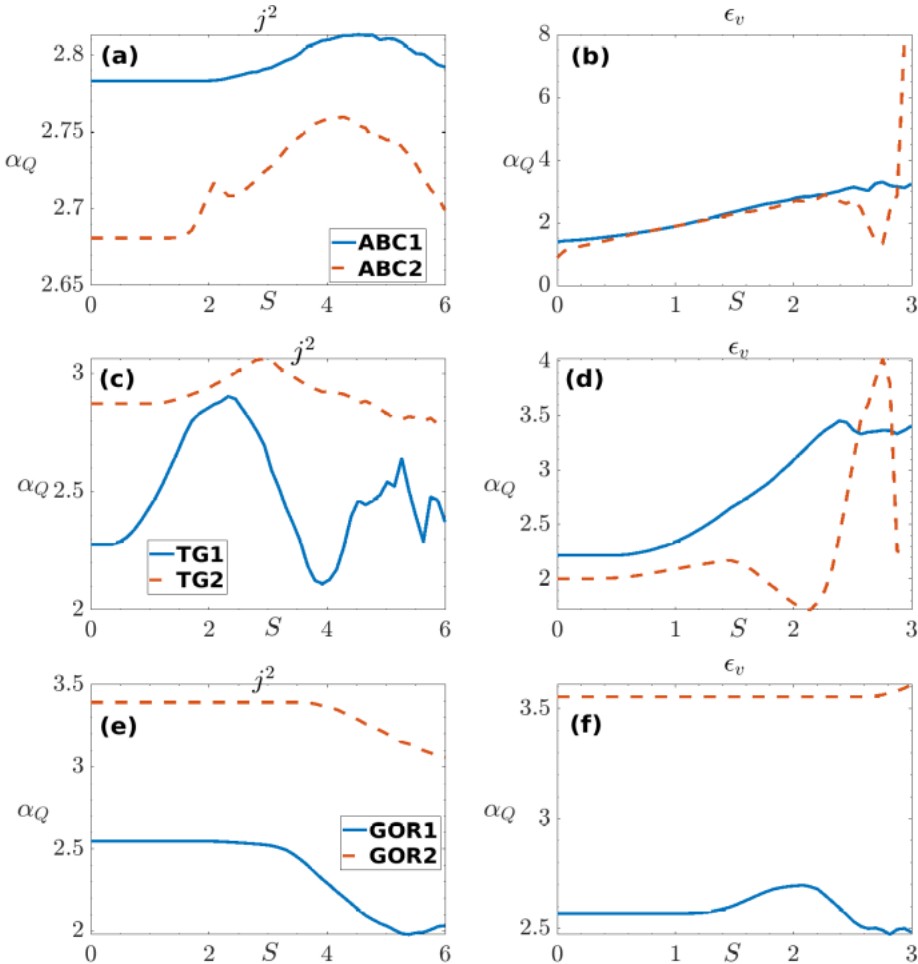

**Figure 4.** Fast dynamos in MHD. Exponents $\alpha$ for the scaling of $Q = \left\langle \delta f^5 \right\rangle / \left\langle \delta f^2 \right\rangle^{5/2}$ as a function of the threshold in skewness, for the magnetic current ($f = j^2$; a, c, e) and kinetic energy dissipation ($f = \epsilon_v$; b, d, f) for runs ABC (a,b), TG (c,d) and GOR (e,f). Note the higher skewness values for $j^2$ when compared to $\epsilon_v$, and its higher dependency on Reynolds number, compared to $\epsilon_v$ for ABC and TG runs.

Plunian (2011) for different runs using the G.O. Roberts dynamo configuration), although the skewness of $j^2$ already reaches high values( above 18).

Preliminary results indicate the following. The energy ratio $r_E = E_M / E_V$ is approaching equipartition (Fig. 3(a)), and the $K(S)$ scatter plot for $j^2$ for run GOR2 in plot (3b) indicates a rather clear power law, the blue line following the parabola $K(S) = 3/2[S^2 - 1]$ (see Garcia (2012)). Note that the color of the dots indicate the time-lapse from the onset of the run, in turn-over times, from blue (early times) to red (late times, of the order of $10^4$, see color bar at right). The remaining graphs of Fig. 3 give scatter plots for the normalized fifth-order moment $Q = \left\langle \delta f^5 \right\rangle / \left\langle \delta f^2 \right\rangle^{5/2}$ for a field $f$ (see equ. (12) below, with

$K_{52} \equiv Q$, in an effort to simplify notations); $f$ is $v_z$ or $b_z$ for plots 3(c,e), and $f$ is $\epsilon_v$ or $j^2$ for plots 3(d,f). In the $Q(S)$ data, power-laws emerge in the tails of the kinetic variables and $b_z$, and throughout for $j^2$.

The scaling exponent $\alpha$ and constant $\kappa$ for the normalized fifth-order moments $Q$ are given in Fig. 4 for the ABC (top), TG (middle), and GOR (bottom) runs for $j^2$ (left) and $\epsilon_v$ (right). They show variations with forcing function, with Reynolds number and with the threshold in $S$ used for the plot, as well as possibly with the equipartition ratio $r_E$. As is the case for the TG and ABC flows, the Reynolds number leads to a difference in scaling for the fit parameters. Thus, runs at higher Reynolds numbers and with several configurations will have to be performed in order to study the scaling of the $\alpha, \kappa$ parameters for relevant variables, but in the next section we begin an approach that can elucidate these scalings laws in the framework of three multifractal intermittency models.

## 4 Theoretical moment scaling using several multifractal intermittency models

### 4.1 Expression for the kurtosis-skewness scaling exponent, $K \sim S^\alpha$, for both fluids and MHD in the SL framework

We now give a path towards a theoretical formulation for $K(S)$ scaling using a classical intermittency model and several of its extensions. Assuming a power-law dependency for velocity and magnetic field structure functions $\delta u(r), \delta b(r)$ defined as

$$\langle [u(x+r) - u(x)]^p \rangle \equiv \langle \delta u(r)^p \rangle \sim r^{\zeta_p^{(f)}} \quad , \quad \langle [b(x+r) - b(x)]^p \rangle \equiv \langle \delta b(r)^p \rangle \sim r^{\zeta_p^{(m)}} \quad , \tag{6}$$

and assuming as well a power-law scaling between kurtosis and skewness for both fluids ($f$) and MHD ($m$), we easily obtain:

$$K_f(S_f) \sim S_f^{\alpha_f} \quad , \quad K_m(S_m) \sim S_m^{\alpha_m} \quad , \quad \alpha_f = \frac{\zeta_4^{(f)} - 2\zeta_2^{(f)}}{\zeta_3^{(f)} - 3\zeta_2^{(f)}/2} \quad , \quad \alpha_m = \frac{\zeta_4^{(m)} - 2\zeta_2^{(m)}}{\zeta_3^{(m)} - 3\zeta_2^{(m)}/2} \quad , \tag{7}$$

with the functions $\zeta_p^{(f,m)}$ depending on the (fluid or MHD) intermittency dynamics or on an explicit model.

We now recall the scaling laws derived in the context of the She-Lévêque formulation for fluids (1994) and for MHD as generalized in Politano et al. (1995). These GSL models, named gslf and gslm respectively for fluids and MHD, depend on two open parameters, $0 < x < 1$ and $0 < \beta < 1$ with, in the limit of the non-intermittent case, $\beta \to 1$, whereas $\beta \to 0$ for a mono-fractal. The anomalous exponents $\zeta_p^{gsl}$ at order $p$ are respectively:[3]

$$\zeta_p^{gslf} = \frac{p(1-x)}{3} + \frac{x(1 - \beta^{p/3})}{1 - \beta} \quad ; \quad \zeta_p^{gslm} = \frac{p(1-x)}{4} + \frac{x(1 - \beta^{p/4})}{1 - \beta} \quad . \tag{8}$$

We note that $x$ is related to the co-dimension of the most dissipative structures in the nonlinear system, and that $\beta$ is a measure of the efficiency of energy transfer and dissipation among intermittent structures as the moment order varies. This formulation leads to log-Poisson statistics (see *e.g.* She and Lévêque (1994); Dubrulle (1994); Frick et al. (1995)). A further assumption of the models concerns the scaling of nonlinear transfer in terms of characteristic times of the problem, namely the nonlinear eddy turn-over time, the wave period (in MHD, the Alfvén wave) and the transfer time of energy to small scales.

---

[3]One could obtain similar relations for the fluid SL model, derived for sheet-like dissipative eddies in Horbury and Balogh (1997). Also, the SL model written in the framework of MHD is generalized in Merrifield et al. (2005) to include extended self-similarity (see also Merrifield et al. (2007) for 2.5D).

The multifractal framework (see Frisch (1995)) allows for a multiplicity of dissipative structures of diverse physical (co)-dimensions: vortex and current sheets, flux tubes, current filaments or bubbles, resulting in a non-integer effective $\beta$ parameter. An extension of the multifractal framework to vectors (velocity field) as opposed to scalars (velocity amplitude), can be found in Schertzer and Tchiguirinskaia (2020). The SL formulation for MHD has been used for example in modeling intermittent nano-flares in connection with solar wind data (Veltri et al., 2005). In the numerical context, it is stressed in Servidio et al. (2011) that a high resolution is needed to quantify properly the properties of local reconnection and current sheets; moreover, reconnection events and the ensuing dissipation are highly local and very varied in amplitude, somewhat reminiscent of the multifractality property reviewed in detail in Lovejoy and Schertzer (2013); Benzi and Toschi (2023).

From equations (8), we can compute the general scaling exponents of kurtosis *vs.* skewness using equation (7). We obtain:

$$\alpha_{gslf} = \frac{2(1 - 2\beta^{2/3} + \beta^{4/3})}{1 + 2\beta - 3\beta^{2/3}} \quad ; \quad \alpha_{gslm} = \frac{2(1 - 2\beta^{1/2} + \beta)}{1 + 2\beta^{3/4} - 3\beta^{1/2}} \quad , \quad \beta \neq 1 \ . \tag{9}$$

Note that, interestingly, both the $\alpha_{gslf}$ and $\alpha_{gslm}$ exponents are independent of $x$, the fractal co-dimension of the most dissipative structures. It is also straightforward to see, again both for fluids and for MHD, that the limit for $\beta \to 0$ is $\alpha \to 2$. In other words, a parabolic law is reached when the most dissipative structure dominates the small-scale dynamics, irrespective of its geometrical (co)-dimension, likely at high $R_V, R_M$ as well as $\langle Ri_g \rangle \approx 1$ (see equations (3)). We note that a similar result on the computation of the $\alpha$ exponent could be written in the context of the model developed in Horbury and Balogh (1997). We also remark however that, in the shell models examined in Frick et al. (1995), $\beta$ never reaches this low limit. Another point concerns the fact that the variation of $\alpha$ could reflect the dependence on the form of the second invariant in the shell models, that is akin to helicity (Frick et al., 1995; Kadanoff et al., 1995). This may point to a limitation of such models when restricted to nearest-neighbor interactions or with different sets of invariants, restrictions that cannot encompass by construction the highly non-local (in scale) interactions leading to anomalous dissipation. Thus, this point will need further investigations.

### 4.2   The standard choice of parameters for the SL models for fluids and MHD

The standard case for the classical fluid SL model is obtained for $x = 2/3$, $\beta = 2/3$ associated with vortex filaments, whereas in the MHD case with wave-vortex interactions and current sheets, the standard parameters become $x = 1/2$, $\beta = 1/2$ (Grauer et al., 1994; Politano and Pouquet, 1995). This yields respectively for the $K(S) \sim S^\alpha$ for fluids ($sslf$) and MHD ($sslm$):

$$\zeta_p^{sslf} = \frac{p}{9} + 2\left[1 - \left(\frac{2}{3}\right)^{p/3}\right] \quad , \quad \alpha_{sslf} \approx 2.56 \ ; \quad \zeta_p^{sslm} = \frac{p}{8} + 1 - \frac{1}{2^{p/4}} \quad , \quad \alpha_{sslm} \approx 2.53 \ . \tag{10}$$

These $\alpha$ values for the standard SL models are also given in Ponty et al. (2025)[4].

All $\alpha$s are close except for extreme cases ($\beta$ at its limits), in part because the values of the anomalous exponents for structure functions for fluids are anchored at $\zeta_3 \equiv 1$. For $\theta, \mathbf{v}$ and $\mathbf{b}$, there are more complex constraints since they involve cross-correlations between fields at third order (Yaglom, 1949; Antonia et al., 1997; Politano and Pouquet, 1998), and also because the analytical expressions for $\alpha$ lead to small fractional power of $\beta$, and we are at relatively low orders of the structure

---

[4]Note that such values are sensitive to the number of decimals taken; in the fluid case using strictly 2 decimals throughout, one finds $\alpha_{sslf} \approx 2.00$.

functions. In fact, an extension of the SL theory to the intermittency of the passive scalar $\theta$ in the fluid case can be found in Lévêque et al. (1999). We can then derive the expression $K_{F_\theta} \sim S_{F_\theta}^{\alpha_\theta}$ in the framework of that model. Here, the scalar flux $F_\theta$ is defined as $F_\theta(r)^{(p)} := \langle |\delta u(r)\delta\theta(r)^2|^{p/3}\rangle \sim \langle |\delta u(r)\delta\theta(r)^2|^{\zeta_p}\rangle$, an expression using the flux arising from the aforementioned exact law for the conservation of scalar energy derived in Yaglom (1949). With the numerical values given in Lévêque et al. (1999), we find $\alpha_\theta^E \approx 2.61$ using anomalous exponents stemming from experiments, $\alpha_\theta^D \approx 2.38$ for DNSs, and $\alpha_\theta^T \approx 2.44$ using the theory developed in that paper. This shows again the sensitivity of these power-laws to the accuracy of the data.

### 4.3 Generalized scaling for higher-order normalized structure functions in the framework of the She-Lévêque models

Let us now rewrite the generalized SL models for fluid and MHD slightly differently, with as before $0 < \beta < 1$ and $0 < x < 1$:

$$3(1-\beta)\zeta_p^{gslf} = x[3(1-\beta^{p/3}) + p(\beta-1)] + p(1-\beta) \ , \ 4(1-\beta)\zeta_p^{gslm} = x[4(1-\beta^{p/4}) + p(\beta-1)] + p(1-\beta) \ . \tag{11}$$

We now compute the scaling of a generalized adimensionalized structure function *vs.* another one, provided they exist, writing:

$$K_{pq} = \frac{\langle \delta u^p\rangle}{\langle \delta u^q\rangle^{p/q}} \ , \ \ K_{rs} = \frac{\langle \delta u^r\rangle}{\langle \delta u^s\rangle^{r/s}} \ , \ \ K_{pq} = f(K_{rs}) = K_{rs}^{\alpha_\sigma} \ , \ \ \sigma = [pr/qs] \ , \ \ \alpha_\sigma = \frac{\zeta_p - [p/q]\zeta_q}{\zeta_r - [r/s]\zeta_s} \ , \tag{12}$$

with $\sigma \in \mathbb{N}^+$, $p > \max[q,r]$, $r > s$. In §4.1, we considered the case $K = S^\alpha$, or in the present notation, $K_{42} = K_{32}^{\alpha_{43/22}}$, with $p = 4, q = 2 = s, r = 3$. After a slightly cumbersome but straightforward computation, one obtains that again $\alpha_\sigma$ is independent of $x$, the co-dimension of dissipative structures, for all values of the indices encapsulated in $\sigma$; one finds specifically:

$$\alpha_\sigma^{(gslf)} = \frac{s}{q}\left[\frac{q(1-\beta^{p/3}) - p(1-\beta^{q/3})}{s(1-\beta^{r/3}) - r(1-\beta^{s/3})}\right] \quad ; \quad \alpha_\sigma^{(gslm)} = \frac{s}{q}\left[\frac{q(1-\beta^{p/4}) - p(1-\beta^{q/4})}{s(1-\beta^{r/4}) - r(1-\beta^{s/4})}\right] \ . \tag{13}$$

In the case of extreme intermittency with $\beta \to 0$, we also have, for both fluids and MHD, and with $s \neq r$ as stated before:

$$\beta \to 0 \ , \ \ \alpha_{pr/qs}^{(gslf),(gslm)} \ \to \ \frac{s(p-q)}{q(r-s)} \ . \tag{14}$$

This formula simplifies, for $q = s$ (same normalization of moments) into $[p-q]/[r-q]$, and gives a parabolic scaling for $p + q = 2r$. Thus, when choosing for the normalisation the second-order energy moment ($q = s = 2$), we have a parabolic scaling for $2r = p + 2$. Similarly, for a normalisation by the skewness, $q = s = 3$, we obtain again a parabola for $2r = p + 3$. These parabolic solutions, for $\beta \to 0$, are directly linked to the algebraic, hierarchical formulations of the SL models.

Finally, let us take two specific examples: $Q(S)$ with $p = 5, q = s = 2, r = 3$, and $H_6(S)$ with $p = 6, q = s = 2, r = 3$ (sometimes called hyper-flatness); the first example for $Q(S)$ is also discussed in Sardeshmukh and Sura (2009). We find in the standard case ($\beta = 2/3$ for fluids and $\beta = 1/2$ for MHD) the scaling $\alpha_{53/22}^{sslf} \approx 4.6$, $\alpha_{53/22}^{sslm} \approx 4.5$, whereas the numerical estimate for the ABC runs gives a maximum of $Q \approx 3.5$. When $\beta \to 0$, $\alpha_{53/22}^{\beta \to 0} \to 3$, a value advocated in Sardeshmukh and Sura (2009) for this $Q(S)$ scaling for both vorticity and potential height using a linear stochastically forced Langevin equation model for climate dynamics with correlated additive and multiplicative noise (see also §5.1 below). To give a second and final example, for $H_6$ in the standard case again, we have $\alpha_{63/22}^{sslf} \approx 7.1$, $\alpha_{63/22}^{sslm} \approx 6.8$, and when $\beta \to 0$, $\alpha_{63/22}^{\beta \to 0} \to 4$, whereas the numerical value we find for the ABC runs is close to 3.7. The discrepancy with the data in Fig. 4 is thus large. In this context, a study in terms of variation with Reynolds number will be informative, but one may have to investigate the MHD turbulence case in 2D, or so-called "2.5D" (two space variations, three components of the fields) to reach substantially higher $R_V, R_M$.

### 4.4 $K \sim S^{\alpha}$ scaling for the Yakhot intermittency model

One can use other models of structure function scaling in turbulent flows. For example, a model of intermittency in fluid turbulence due to Yakhot (2006) (herewith model Y) yields the scaling:

$$\zeta_{2p}^{(Y)} = \frac{2(1+3\beta_y)p}{3(1+2p\beta_y)} \ , \ \ \zeta_3^{(Y)} = 1 \ \forall \ \beta_y \ ; \ \ with \ q = 2p \ , \ \ \zeta_q^{(Y)} = \frac{q(1+3\beta_y)}{3(1+q\beta_y)} \ . \tag{15}$$

The model comes from evaluating perturbatively the corrections to two-dimensional turbulence when close to a critical dimension at which the energy cascade reverses its direction to the small scales. One can verify that $\beta_y = 0$ gives a $\zeta_p = p/3$ standard scaling. We immediately get $\alpha_y \approx 2.56$ for the relationship $K \sim S^{\alpha_y}$ when choosing for the open parameter the value $\beta_y \approx 0.05$ close to that given by experiments (see also Nickelsen (2017); Friedrich and Grauer (2020) for recent analyses of this and other models[5]). The anomalous $\zeta_p$ exponents themselves (see Fig. 1 in Friedrich and Grauer (2020)) do not differ by much from model to model, specially at relatively low order. But in view of the sensitivity of $\alpha$ to the evaluation of the anomalous exponents, $\alpha$-scaling in an empirical $K(S)$ law may prove a valuable tool in order to discern between different intermittency modeling and small-scale parametrisation in general, somewhat better than with the $\zeta_p$ themselves, given sufficiently resolved data leading to precise fits to the quasi-parabolic power-law behavior for long runs in terms of turn-over times.

In conclusion, if the change of $K(S)$ scaling with Reynolds number is not known, and is difficult to evaluate experimentally or numerically, the data is sufficient to assess that such scalings will be observed at high $R_V$; indeed, it can be expected in the framework of random multiplicative systems (see Benzi and Toschi (2023) for a recent introduction). We also recall here that a parabolic law can be justified on several grounds. First, as stated earlier, one can write a Taylor expansion for $K(S)$ for a quasi-Gaussian PDF, and note that $K \geq 0$. Another reason for observing a $K(S)$ parabolic law relies on the existence of Cauchy-Schwarz relationships (and their generalisations) between the third- and fourth-order moments of a stochastic variable $f$, namely $S_f^2 \leq K_f + 3$, with a tightening of the inequality for a unimodal PDF, namely $S_f^2 \leq K_f + 186/125$ for a finite fourth-order moment (Klaassen et al., 2000). We also note that *Beta* distributions are advocated in Labit et al. (2007) for the intermittency of density fluctuations in drift-exchange turbulence in plasmas, in particular because they admit both positive and negative skewness as observed in many instances such as the fast dynamo (Ponty et al., 2025). These $K(S)$ relations also provide useful bounds for the data.

## 5 Other approaches for quasi-parabolic scaling beyond the GSL and Y models

### 5.1 Linear and non-linear Langevin models

Langevin equations have long been written in the context of turbulent flows, for example in order to take into account the nonlocality of mode interactions leading to intermittency, modeling as such the separation of spatial and temporal scales (Nazarenko et al., 2000a; Laval et al., 2003). Indeed, dissipative intermittent structures such as shear layers or current sheets are multi-scale,

---

[5]In the Markov process ($M$) interpretation of the SL model, the independence of $\alpha$ on the co-dimension of dissipative structures is an independence of the jump distribution on the associated stochastic process due to $M$, and only the amplitude of the velocity jumps (leading to dissipation intermittency) matters.

spanning a range from the integral scale characteristic of their length to the dissipative scale defined by viscosity or resistivity, *e.g.* the Kolmogorov scale $\eta_K$ for NS, providing one way intermittency can be found both at large and at small scales. We note however that, in the multifractal framework such as in the SL models, there is a range of dissipative scales corresponding as well to a plage of spectral indices. This provides a justification for the application of a Langevin framework, where the original nonlinearities of the primitive equations are modeled through fast-evolving additive and multiplicative stochastic noise. It is shown in Wan et al. (2012) that the kurtosis of the magnetic field filtered at the dissipation scale and smaller increases sharply and significantly both in high-resolution 2D DNS and in ACE and Cluster solar wind data. Recent observations in the heliosphere analyzing data from the Parker Solar Probe confirm the importance of such non-local interactions in the case of so-called imbalanced MHD turbulence with $\mathbf{z}^{\pm} = \mathbf{v} \pm \mathbf{b}$ of unequal amplitudes (Yang et al., 2023), an imbalance enhanced by the quasi-absence of collisions (Miloshevich et al., 2021).

One can write a stochastic Langevin equation for a fluctuating field $\tilde{c}$, *viz.* $D_t \tilde{c} = -(\bar{\lambda}_k + \lambda'_k)\tilde{c} + \tilde{\zeta}_k$, where $[\bar{\lambda}_k, \lambda'_k]$ represent large-scale and fluctuating small-scale velocity stretching the magnetic field lines in the kinematic phase, and $\tilde{\zeta}_k$ is an additive noise due to (plausible) rapid small-scale fluctuations. The essential features in the development of Sura and Sardeshmukh (2008) for climate can thus be reproduced in the MHD case; this will likely lead to the same conclusion of a parabolic behavior. The large-scale velocity and induction are constrained by divergence-free conditions, by Galilean invariance for the velocity, and perhaps even more importantly by existing so-called exact laws[6]. Such laws involve third-order cross-correlations of $\mathbf{u}$ and $\mathbf{b}$ (see Marino and Sorriso-Valvo (2023) for a recent review), whereas the fourth-order moments do not have such constraints for quadratically nonlinear equations. A non-zero energy dissipation rate (a plausible conjecture) thus implies non-Gaussianity ($S \neq 0, K \neq 0$). A Langevin equation developed in the kinematic dynamo regime can be amended to model the back reaction of the Lorenz force, as discussed briefly in Ponty et al. (2025). We finally note that, starting from well-resolved data, one can reconstruct a Langevin equation model of the observed stochastic process (Friedrich et al., 2011; Rinn et al., 2016). This may prove instructive, in particular if different models were to emerge for different regimes or dynamo types.

## 5.2 The nonlinear Langevin approach for the dynamo regime

Several Langevin approaches in MHD have been derived in the nonlinear case (see Zwanzig (1973) for an early study for fluids). For example, a sub-diffusive behavior was unraveled in Balescu et al. (1994), from first principles, in the context of a stochastic magnetic field. One can also choose to add a cubic term in the induction equation (cubic so that the symmetry of the axial magnetic field be preserved), in order to mimic the effect the Lorentz force has on the velocity (see *e.g.* Boldyrev (2001); Leprovost and Dubrulle (2005)), in particular for large magnetic Prandtl numbers. On the other hand, it was shown in Nazarenko et al. (2000b) that in the case of the fast dynamo, the feed back of the growing induction is through the creation of counter-rotating vortices, a point not included in a saturation involving only the magnetic field equation. One can also consider the role of Alfvén waves in the nonlinear regime by introducing in a linear Langevin equation an oscillatory term (Bandyopadhyay et al. (2018)). Note that, in a Langevin equation, in a sense, one is getting rid of the closure problem for

---

[6]These exact laws have been extended to fluid and MHD turbulence as apply to the heliosphere, see *e.g.* Ferrand et al. (2021); David and Galtier (2022).

turbulent motions since it is linear, with the complex nonlinear small-scale dynamics bundled up in a rapid stochastic forcing with an assumption of (mostly) local interactions among these fast motions.

## 5.3 Self-organized criticality and $1/f$ law as another possible framework for intermittent quasi-parabolic scaling

Self-Organized Criticality (SOC) has been introduced in the context of sandpile systems and their avalanching properties (Bak et al., 1987), as when modelling solar flares (Lu and Hamilton (1991); see also Bramwell et al. (2000); Chapman and Watkins (2001); Osman et al. (2014); Watkins et al. (2016); Balasis et al. (2023) for recent discussions). It can be seen as a system with slow driving and fast relaxation, leading to power-law scaling of spatio-temporal dissipative avalanches. In the context of DNS in three-dimensional (3D) MHD, Uritsky et al. (2010) identified SOC in the dissipative range of decaying runs (so with a local critical Reynolds number of order unity). However, SOC was not found in the inertial range, a fact that was interpreted as SOC properties propagating from the dissipative to the inertial range, with merging of current structures. Indeed, the dissipative features of turbulent flows are multiscale, spanning from the energy containing range to the dissipative one, such as in vortex filaments and current sheets (see Watkins et al. (2016) for a comparative study of definitions for SOC behavior). The critical state is that in which the source (the energy cascade at a fixed rate) and the sink (the dissipation at a fixed rate through *e.g.* eddy viscosity) balance, as they do on average. Note that Smyth et al. (2019) identified SOC in rotating stratified flows with the Richardson number, governing shear instabilities such as KH, being the critical parameter (see also Fig. 2). The nonlinear interactions in the inertial range are conservative, and dissipation sets in through nonlocal interactions between energy-containing eddies and dissipative ones, lending these interactions to be described by SOC together with $1/f$ noise (Vespignani and Zapperi, 1998). As shown in Dmitruk and Matthaeus (2007), this leads to an emphasis on the dynamics of the largest modes, and on their interactions with the early dissipative range where intermittency is strongest (Kraichnan, 1967b; Chen et al., 1993). Also, the sharp variations of the flow and field due to the nonlinearities of the primitive equations can be treated as a stochastic force using renormalization group techniques, reminiscent again of a Langevin approach (Materassi and Consolini, 2008). In all these studies, nonlinear shear instabilities appear central to the inter-related small-scale and large-scale behavior of the stochastic turbulent flows.

## 6 Conclusion and perspectives

We have analyzed in this paper the relative behavior of normalized moments of the velocity, magnetic field, as well as temperature fluctuations, in a variety of contexts, and we have given a rationale to cast these results in the mold of classical intermittency models for fluid and MHD turbulence, models which provide a natural framework for such relative scalings. The variability of the scaling is linked to the details of the dissipative structures and their relative intensities. The ubiquity of a quasi-parabolic $K(S) \sim S^\alpha$ law could be interpreted as it having no specific physical meaning; on the other hand, it may be pointing to a universality of intermittency in turbulent flows. We also note that the power-law exponent $\alpha$ is independent of the (co)-dimension of the dissipative structures. The abrupt transition in $\alpha$-scaling for the rotating-stratified case when shear instabilities arise (see Fig. 2) is indicative of an underlying dynamics where the development of turbulence, as measured by the

Ozmidov scale becoming larger than the dissipative scale in that case, plays a dynamical role (Pouquet et al., 2023). In MHD, one issue absent from the present analysis is to incorporate the potential effect of helical structures (with non-zero kinetic, magnetic and/or cross helicity) on the $K(S)$ scaling. It is known from multiple studies that helicity plays a central role in large-scale dynamos (see Brandenburg and Subramanian (2005)), and that its incorporation in closures of turbulence leads to better modeling of these flows (Yokoi, 2013).

The multifractality of the She-Lévêque model is measured through the $\beta$ parameter. As $\beta \to 0$, the intermittency of the flow is carried by one single structure and the flow becomes, in that extreme case, monofractal. In such a limit, $\alpha \to 2$ (see equ. (9)), so that a strict parabolic behavior for $K(S)$, in the framework of such models, is linked to monofractality. We also note that PDFs of the potential energy dissipation rate computed in this paper for quasi-geostrophic flows have exponential tails (see Fig. 2), with lesser decay as the Reynolds number is increased. For the so-called $\alpha-$stable processes, the scaling exponents of normalized moments of the multifractal dynamics can be computed, and it can be associated with (in some cases, unbounded) singularities (see Serinaldi (2010) in the context of rainfall). Furthermore, such multi-fractal analysis could give information on latitudinal dependency of data, and unravel different regimes in the dynamics such as, in the atmosphere, the weather, macroweather and climate (Lovejoy, 2018).

In this context, a standard analysis of anomalous exponents of temporal structure functions (as opposed to spatial ones) on the various data sets presented herein will be of great interest and is planned for future work. Moreover, temporal moment analysis allows one to sort out shorter and longer timescales in time series of nonlinear phenomena, and their statistics such as in climate data (Franzke et al., 2020). Finally, multifractal analysis, through the computation of anomalous scaling exponents of order $p$, and of an evaluation of their limit as the order $p \to \infty$ (see, *e.g.* Pierrehumbert (1996)), can lead to useful characterizations of turbulent structures, as for example in $km$-size clouds (Freischem et al., 2024). Add something?

In order to pursue the investigation of $K(S)$ laws in turbulence at higher Taylor Reynolds number, one can implement hyper-viscosity algorithms, or else use models which, because they are significantly less costly numerically, will allow for longer statistics at substantially higher $R_V$. Such approaches are numerous. One can think of shell models retaining only one mode per field per wavenumber shell and only nearest-neighbor interactions as developed for MHD in Gloaguen et al. (1985) (see Plunian et al. (2013) for review). One can also simplify the dynamics by lowering the space dimension, as for the 1D, 2D and 2.5D cases (see *e.g.* Thomas (1970); Hada (1993); Laveder et al. (2013); Merrifield et al. (2007); Servidio et al. (2011)). Numerical adaptation, preferably spectral when dealing with $L_\infty$ norms as for extreme intermittent events (see Ng et al. (2008)), as well as various large-eddy simulations (Sagaut and Cambon (2008)), or the so-called $\alpha$-model (Holm et al., 1998) used fort example in the framework of oceanic dynamics (Pietarila Graham and Ringler, 2013) or analyzed as well in MHD (Montgomery and Pouquet, 2002) will be similarly useful. These methods will allow for disentangling between Reynolds number and intermittency effects, the consequences of the presence or not of helicity linked to vortex filaments and to the dynamo, as well as equipartition or not of kinetic, potential or magnetic energy.

One further important issue will concern incorporating the role of anisotropy which can affect scaling properties and interpretations of the intermittency, as shown in the context of the atmosphere in Lovejoy et al. (2001), or in Schekochihin (2022) for MHD. Finally, it was noted in Yeung et al. (2018) that the grid resolution, Courant number and machine precision all affect

the estimate of the overall enstrophy. Furthermore, the scaling for the strongest gradients becomes linear in $R_\lambda$ at high values of $R_\lambda$, with intermittent structures found at scales smaller than the Kolmogorov scale $\eta_K$ (Buaria et al., 2019), confirming the existence of intermittency beyond $\eta_K$. In addition, in Buaria and Pumir (2024), a relative scaling of moments of the velocity

gradient tensor (restricted to longitudinal components) is analyzed using high $R_\lambda$ numerical and experimental fluid turbulence data. These authors show the possibility of a universal scaling behavior of relative moments where $R_\lambda$ disappears, and with explicit data on $H_6$. This type of analysis is not performed here for lack of sufficiently large Reynolds number, and the ensuing lack of sufficiently intense localized dissipation, but a study of intermittent structures in MHD at substantially higher Reynolds number is planned for the future.

*Code and data availability.* Codes and data are available upon reasonable request.

*Author contributions.* All authors contributed equally to this work.

*Competing interests.* No competing interests are present.

*Acknowledgements.* This paper is written in the context of the 2024 Lewis Fry Richardson medal of the Nonlinear Geophysics section at EGU, to which Annick Pouquet is very grateful. She also wants to thank the many mentors, collaborators, students and post-doctoral fellows

with whom she interacted over many years, principally in France (Nice, Paris, Lyon) and the US. Of particular mention are Axel Brandenburg, Robert Ergun, and Stuart Patterson on both sides of her career, as well as Pablo Mininni and the co-authors of this paper.
Raffaele Marino acknowledges support from the project "EVENTFUL" (ANR-20-CE30-0011), funded by the "Agence nationale de la recherche" through the program AAPG-2020. Yannick Ponty thanks A. Miniussi for computing design assistance on the CUBBY code. The authors are grateful to the OPAL infrastructure from Université Côte d'Azur, the Université Côte d'Azur's Center for High-Performance

Computing, PMCS2I at the École Centrale de Lyon and to the national French computer facilities (GENCI) for providing resources and support. Duane Rosenberg acknowledges support from award NOAA/OAR/NA19OAR4320073. NCAR is funded in part by NSF. Finally, we are thankful to both referees for useful remarks.

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
