# Peer review of "Intermittency in fluid and MHD turbulence analyzed through the prism of moment scaling predictions of multifractal models"

_EGUsphere, 2024_

## Referee Comment (RC2)

**Review of the manuscript « Intermittency in fluid and MHD turbulence analyzed through the prism of moment scaling predictions of multifractal models » by Pouquet et al.**

This manuscript strongly focuses on the "quasi-parabolic" K(S) law relating kurtosis (K) to skewness (S) by a power law with an exponent close to 2 for a wide variety of turbulent fields ranging from fluid flows to MHD and beyond. In particular, the authors argue that the She-Lévêque (SL) model, including its generalization for MHD, yields a parabolic law K(S) for maximum intermittency.

The fact that this manuscript was prepared as part of the 2024 EGU Lewis Fry Richardson Medal and is co-authored with leading turbulence specialists reinforces NPG's interest in publishing this paper as is, being considered a legacy. However, I believe that this paper can be further improved, at the discretion of the authors. I hope the comments and suggestions below could be helpful in this direction.

- Page 1 (L10-16):

It is surprising that in their introduction to intermittency, the authors do not really mention the wind turbulence that has been considered *per se* for some time, at least since Kolmogorov 1941 and the questions raised by Arnold about its intermittency, which led to first models of intermittency (Kolmogorov, 1962). Furthermore, fundamental paradigm shifts occurred when two earlier EGU Lewis Fry Richardson Medalists demonstrated that the atmospheric spectrum could no longer be split into 2D/3D turbulence for large and small scales respectively (e.g., Lovejoy and Schertzer (2013)).

- Page 1 (L38):

It could be restrictive to define intermittency by the presence of "localised structures", whereas if structures are more obvious at small scales, the scaling of turbulence nevertheless implies structures at all scales.

- Page 3 (L81-83):

Contrary to what is stated on L82, the kurtosis is not "recentered" by 3 in what follows. It may be useful to recall that mono/uni-fractality of the field $f$ yield strictly scale invariant $K$ and $S$, which is no longer the case for multifractal fields.

- Page 4 (L84-91):

It would be interesting to have early historical overview of the quasi-parabolic relation between the skewness and the kurtosis

- Page 4 (L99 and elsewhere, including Fig.1 caption):

$\varepsilon_v$ is presented as the kinetic energy dissipation, whereas it could correspond to the energy flux density. It should be made very clear from which papers the figures of Fig.1 are taken should be clearly indicated. This is a systematic problem of every figure caption.

- Section 4.1 (L.162-171):

- the She-Lévêque (SL) model is the main model used to theoretically derive a few properties of the $K(S)$ law, but it is only abruptly presented by the scaling exponent $\zeta_P$ of its structure function (Eq.7), not the stochastic model itself;

- whereas important limitations may result from the fact that, like shell models (e.g. Frick et al., 1995), SL has only scales, not any space extension, and hence properly speaking no structures, contrary to multiplicative cascade models and, of course, DNS;

- in particular the $x$ parameter (often noted $\Delta$ in other papers) is supposed, as usual, to be related to the co-dimension of the most dissipative structures, but this is not obvious due to the previous remark, however this parameter does not play a key role in what follows;

- the role of the $\beta$ parameter is much more important (see Eq. 7). Its definition does not seem fully satisfactory for me, but could be understood as the sensitivity of the intermittency to the statistical order. The main problem that its value 0 is from time to time claimed to correspond to maximal intermittency (e.g. in the abstract and L.153-154), while elsewhere it is indeed regarded as vanishing intermittency (e.g. L.165).

- Section 5

Both section 5.1 (Langevin models) and 5.2 (S.O.C.) only seem to describe other approaches, but do not yet provide quantitative results. Regarding S.O.C., it seems surprising that P. Bak is not cited.

- Section 6 (Conclusion)

In their conclusions, the authors draw up an impressive list of research extensions, e.g. taking helicity into account. However, to clarify the possible outcome of the present work, I would suggest that the authors look at the framework of universal multifractals (see the references already cited), in particular to show that the strict parabolic law of K(S) is reached with vanishing multifractality. This could be particularly helpful for understanding the physics behind a quasi-parabolic law $K(S)$ for maximum intermittency.

---

## Author Response (AR1)

**Intermittency in fluid and MHD turbulence analyzed through the prism of moment scaling predictions of multifractal models**

Annick Pouquet[1], Raffaele Marino[2], Hélène Politano[3], Yannick Ponty[4], and Duane Rosenberg[5]

[1]National Center for Atmospheric Research, P.O. Box 3000, Boulder, CO 80307, USA
[2]Laboratoire de Mécanique des Fluides et d'Acoustique, CNRS, École Centrale de Lyon, Université Claude Bernard Lyon 1, INSA de Lyon, F-69134 Écully, France
[3]Université Côte d'Azur, CNRS, LJAD, France
[4]Université Côte d'Azur, CNRS, Observatoire de la Côte d'Azur, Laboratoire Lagrange, France
[5]Cooperative Institute for Research in the Atmosphere (CIRA), NOAA/OAR Global Systems Laboratory, Colorado State University, 325 Broadway Boulder, Fort Collins, CO 80305, USA

**Correspondence:** Annick Pouquet (pouquet@ucar.edu)

**[RESPONSE TO REFEREES]**

We give below responses to remarks and comments made by the two reviewers through the NPG channel. In this document, the texts of the reviewers are given, followed by our responses (in blue) for each point. In the text of the manuscript, major changes are indicated in blue as well, for clarity.

We have also changed a paragraph at the very end of the conclusion; in particular, we had a wrong reference which we now corrected. Finally, note that not all references added to the text as per the requests of both referees are at the end of this letter.

We thank both referees for useful remarks which have made the paper more clear and complete. We have also changed the bibliography style following the web-of-science Cornell data base, as per editorial instructions.

We feel we have positively answered their remarks, and we now we resubmit our paper to NPG.

Annick Pouquet
On behalf of all authors

Boulder, March 5, 2025

.../...

**1 Response to the first reviewer**

**1.** "This article is an interesting study on the detection of intermittency in different fluids, which is a vast subject. After an excellent introduction with a review, the article presents a new study on the relationship between kurtosis and skewness. The authors discuss a possible parabolic relationship between these two quantities. This relation is based on direct numerical simulations at moderate resolution. Although the study is considered preliminary, it is already interesting and I think it can be published in NPG. I have a few comments which I list below. "

*Our response:* We thank the reviewer for these positive remarks.

**2. Comments:**

"The abstract begins with the assertion that in the presence of waves, certain characteristics remain in turbulence, such as intermittency. I think that when the amplitude of these waves is weak, we generally do not find intermittency. Am I wrong? Please add a comment. "

*Our response:* The reviewer is right, we are mostly stressing in this paper the intermediate regime where waves and nonlinear eddies are in interaction and intermittency is strong. We added comments in the abstract and in the core of the text.

**3.** "After equation (2), I think the total pressure equation is not written correctly. The magnetic field/pressure should appear. Is it incompressible?"

*Our response*: A mistake, now corrected, with the total pressure properly defined as $P = p + |\mathbf{b}|^2/2$. In the MHD case, the flow is also assumed here to be incompressible, and this is now stated in the text.

**4.** " I am having trouble reading Fig. 2, especially the abscissa. And the file name at the top seems useless. Please improve it. "

*Our response:* All figures have been examined carefully, and rebuilt when necessary.

**5.** "Figure 3 is even harder to follow. I suggest adding references (a, b, c, d...) and using these names to comment on the results. In addition, it is sometimes difficult to read the information, as in the bottom left-hand figure. This is important because the discussion is mostly focused on it. "

*Our response:* See the response to the point above. We have also decided that Figure 3 was too complex and we have both split it into Figures 3 and 4, and simplified it further by suppressing some of the plots (removing those concerning $H_6$ with no incidence on the rest of the paper).
We have also named each plot (a,b,c,d, ...) as appropriate, and we have changed the text of the paper accordingly. We think indeed that the paper is more clear with these changes.

**6.** "Line 128: where can we find the definition of Q5?"

*Our response:* Q5 (renamed Q) is now defined explicitly in the text around that line, and the reference to equation (12) is made more clear. The text around equation (12) is also made a bit more explicit. We note that we have attempted to have clear explicit notations for complex but straightforward expressions of normalized moments of any order, to show the underlying simplicity and unity of these concepts/results.

**7.** "A comparison is made (page 8) with papers on MHD. I would suggest also comparing with Horbury & Balogh (1997) where a log-Poisson distribution is proposed. Is it possible to make a comparison?"

*Our response:* We have added a line on the derivation in Horbury & Balogh (1997) of the extension of the She-Levêque model to sheets as the main dissipative structure (see also footnote 4). We note that we already mentioned in the original version of the manuscript the log-Poisson character of PDFs in that framework.

We have not added a paragraph on the evaluation of the $\alpha$ exponent(s) in the framework of the SL model with dissipative structures in the form of sheets as proposed in Horbury and Balogh (1997) (corresponding to $x = 1$) in order not to burden the paper which already contains quite a lot of detailed formulæ. Of course, the reason for this choice is that, in the physically relevant limit of high Reynolds number, the $\alpha$ exponents are seen, interestingly enough, *not* to depend on the (co)-dimension of the dissipative structures. We also feel the data is not sufficient to actually display departures from these different scalings, and between scaling laws, the point of the paper being that in fact different scaling can occur for different dissipative structures. As for determining the functional shape of Probability distribution functions (PDFs), this is notoriously difficult since the wings are determinant and yet carry the most uncertainty. In the quasi-geostrophic stratified case (see Figure 2), PDFs are displayed and we give in the text an exponential fit. Higher Reynolds number computations for long times have to be performed, as mentioned in the concluding remarks.

**8.** "In conclusion, I find the paper interesting and after these small improvements, I think it can be published. I would also like to congratulate the first author, as I see that the article is written in the context of the Lewis Fry Richardson 2024 medal. So congratulations for your wonderful contribution to turbulence! "

*Our response:* We are thankful to the reviewer for this positive assessment of our paper, and also are grateful for the recognition that such a prestigious price brings to all of the authors, and the community of geophysical turbulence at large.

**2 Response to the second reviewer**

**1.** "This manuscript strongly focuses on the "quasi-parabolic" K(S) law relating kurtosis (K) to skewness (S) by a power law with an exponent close to 2 for a wide variety of turbulent fields ranging from fluid flows to MHD and beyond. In particular, the authors argue that the She- Lévêque (SL) model, including its generalization for MHD, yields a parabolic law K(S) for maximum intermittency.

The fact that this manuscript was prepared as part of the 2024 EGU Lewis Fry Richardson Medal and is co-authored with leading turbulence specialists reinforces NPG's interest in publishing this paper as is, being considered a legacy. However, I believe that this paper can be further improved, at the discretion of the authors. I hope the comments and suggestions below could be helpful in this direction."

*Our response:* We thank the reviewer for this assessment of our work, and we want to tackle the reviewer's suggestions for changes to the manuscript in the following.

**2.** Page 1 (L10-16): It is surprising that in their introduction to intermittency, the authors do not really mention the wind turbulence that has been considered per se for some time, at least since Kolmogorov 1941 and the questions raised by Arnold about its intermittency, which led to first models of intermittency (Kolmogorov, 1962). Furthermore, fundamental paradigm shifts occurred when two earlier EGU Lewis Fry Richardson Medalists demonstrated that the atmospheric spectrum could no longer be split into 2D/3D turbulence for large and small scales respectively (e.g., Lovejoy and Schertzer (2013)).

*Our response:* We have now added two full paragraphs on a few of the foundational concepts and observations on intermittency in turbulent flows at the beginning of §3. Indeed the paradigm shift mentioned by the reviewer, stressing among others the importance of anisotropy to reveal an overall structure among scales, is important but not directly related to what we do in this paper.

**3.** Page 1 (L38): It could be restrictive to define intermittency by the presence of "localized structures", whereas if structures are more obvious at small scales, the scaling of turbulence nevertheless implies structures at all scales.

*Our response:* The reviewer is right, due to the ambiguity associated with localization; but strong small-scale structures are perhaps more easily detectable. Another important property of intermittency is the anisotropy of such structures, as already noted in our manuscript, linking small and large scales in a highly non-local way (see the beginning of §3, as well as §5.1 in which we added a sentence).

**4.** Page 3 (L81-83): Contrary to what is stated on L82, the kurtosis is not "recentered" by 3 in what follows. It may be useful to recall that mono /uni-fractality of the field f yield strictly scale invariant K and S, which is no longer the case for multifractal fields.

*Our response:* The equation on page 3 (below equation 4) is simply a definition so that for the Gaussian case, (excess) kurtosis is zero, as the skewness; it is now labeled equation (5).

**5.** Page 4 (L84-91): It would be interesting to have early historical overview of the quasi-parabolic relation between the skewness and the kurtosis

100    *Our response:* We note that the earlier studies of $K(S)$ quasi-parabolic laws were reviewed in the first version of the manuscript separately for fluids (around lines 90, beginning of §3.1) and for MHD ($\approx$ lines 130 *sq.*, second paragraph of §3.2). We have added two paragraphs at the beginning of §3.1 reviewing some of the classical studies on $K(S)$ laws.

**6.** Page 4 (L99 and elsewhere, including Fig.1 caption): $\epsilon_v$ is presented as the kinetic energy dissipation, whereas it could correspond to the energy flux density. It should be made very clear from which papers the figures of Fig.1 are taken should be 105  clearly indicated. This is a systematic problem of every figure caption.

   *Our response:* $\epsilon_v$ is classically called energy dissipation, and we shall not change this nomenclature.

Also, all figures in the manuscript are original, and there are also new runs using the Roberts flow for the dynamo case.

As for Figure 1, it is an original figure with all new unpublished plots, as are those in Figures 2 and 3, and now Figure 4, stemming from the severing of the last figure of the original manuscript into two separate figuires, for clarity of exposition.

110  **7.** Section 4.1 (L.162-171): the She-Lévêque (SL) model is the main model used to theoretically derive a few properties of the K(S) law, but it is only abruptly presented by the scaling exponent $\zeta_p$ of its structure function (Eq.7), not the stochastic model itself whereas important limitations may result from the fact that, like shell models (e.g. Frick et al., 1995), SL has only scales, not any space extension, and hence properly speaking no structures, contrary to multiplicative cascade models and, of course, DNS; - in particular the $x$ parameter (often noted $\Delta$ in other papers) is supposed, as usual, to be related to the co-dimension of 115  the most dissipative structures, but this is not obvious due to the previous remark, however this parameter does not play a key role in what follows;

   *Our response:* We do not feel the need to expose the SL model in detail, our paper being already quite long. The reviewer is absolutely right that the SL models are quite limited in the sense that they do not have spatial information nor do they have anisotropy, the $x$ (or $\Delta$) parameter resulting from a greatly simplified modeling of physical structures. But it has nevertheless 120  proven to be powerful and useful. due to its simplicity in structure. Further studies will need to be done to take into account the geometry of dissipative elements, but this is beyond the scope of this paper. As for the $x$ parameter, indeed it disappears from the formulation, and we feel that is a perhaps unexpected but an interesting new result. Incidentally, the $x$ notation originated in the Politano and Pouquet (1995) paper and we decided to keep it to make the referencing to that paper simpler.

**8.** - the role of the $\beta$ parameter is much more important (see Eq. 7). Its definition does not seem fully satisfactory for me, but 125  could be understood as the sensitivity of the intermittency to the statistical order. The main problem that its value 0 is from time to time claimed to correspond to maximal intermittency (e.g. in the abstract and L.153-154), while elsewhere it is indeed regarded as vanishing intermittency (e.g. L.165).

   *Our response:* We thank the reviewer for pointing to a somewhat ambiguous writing in the original manuscript which we have now corrected; $\beta = 1$ corresponds to the non-intermittent case and $\beta = 0$ to the case when one single structure dominates 130  the (extreme) intermittency.

**9.** Section 5 Both section 5.1 (Langevin models) and 5.2 (S.O.C.) only seem to describe other approaches, but do not yet provide quantitative results. Regarding S.O.C., it seems surprising that P. Bak is not cited.

*Our response:* We wanted to provide a short reminder of other possible approaches, which are left to be tackled in future works (but see the interesting derivation in the climate context in Sardeshmukh and Sura (2009) for a Langevin approach). Nevertheless, we agree with the reviewer that a citation to P. Bak is certainly needed, as is one to Arnold. They are now included, referencing these classical papers: Bak et al. (1987); Arnold (1963) (see also Arnold and Khesin (2021)).

**10.** Section 6 (Conclusion) In their conclusions, the authors draw up an impressive list of research extensions, e.g. taking helicity into account. However, to clarify the possible outcome of the present work, I would suggest that the authors look at the framework of universal multifractals (see the references already cited), in particular to show that the strict parabolic law of K(S) is reached with vanishing multifractality. This could be particularly helpful for understanding the physics behind a quasi-parabolic law K(S) for maximum intermittency.

*Our response:* A multifractal framework study certainly needs to be performed further in depth, but this is again left for future work. Indeed, we agree that the fact that maximum intermittency leads to a parabolic $K(S)$ law is of interest and its consequences are worth pursuing. We note *en passant* that a remark on the secondary role in intermittent scaling of the dimension of dissipative structures was already made in fact in Dubrulle (1994) in a different context, as already cited in the previous version of the paper. Here, in a sense, we find it using a specific model and we develop some of the consequences of this fact for the evaluation of specific relationships between a variety of normalized moments of different orders.

In conclusion, we thank the reviewer for these comments and we think they have helped improve the paper.

.../...

**3 On one remark received directly by mail**

Dhawal Buaria and Alain Pumir remarked to us, accessing our paper on the web, that the following passage towards the end of the conclusion, marked in green below, was inaccurate; it is now replaced by the text marked in blue, with the text in black giving the context of the paragraph shortly before and after the incriminated sentences, *viz.*:

**OUR TEXT:** ... One further important issue will concern incorporating the role of anisotropy which can affect scaling properties and interpretations of the intermittency, as shown in the context of the atmosphere in Lovejoy et al. (2001), or in Schekochihin (2022) for MHD.

**DELETE:** It was shown in Yeung et al. (2015) that for extreme events, defined as having their local dissipation being more than $10^4$ above the mean, the strongly intermittent vorticity structure is a sub-part of the usual vortex filament and appears more as an (isotropic) blob; thus, these intense structures are not force-free (which would require, for the NS case, quasi-parallel velocity and vorticity as in a filament) and are therefore dissipative.

**REPLACE BY:** Finally, it was noted in Yeung et al. (2018) that the grid resolution, Courant number and machine precision all affect the estimate of the overall enstrophy. Furthermore, the scaling for the strongest gradients becomes linear in $R_\lambda$ at high values of $R_\lambda$, with intermittent structures found at scales smaller than the Kolmogorov scale $\eta_K$ (Buaria et al., 2019), confirming the existence of intermittency beyond $\eta_K$.

**OUR TEXT AGAIN:** This type of analysis is not performed here for lack of sufficiently large Reynolds number in our data, and the ensuing lack of sufficiently intense localized dissipation, but a study of intermittent structures in MHD at substantially higher Reynolds number is planned for the future.

*Nota-1: Consequently, the reference Yeung et al. (2015) has been removed, and references Yeung et al. (2018); Buaria et al. (2019) have been added instead.*

*Nota-2- Not all references added to the text are listed on the next page; similarly, not all references in the next page are new to the manuscript.*

.../...

**References**

175   Arnold, V.: Proof of a theorem of A.N. Kolmogorov on the invariance of quasi-periodic motions under small perturbations of the Hamiltonian, Rus. Math. Surv+, 18, 9–36, 1963.

Arnold, V. and Khesin, B.: Topological Methods in Hydrodynamics, Second Edition, Springer-Verlag, New York, 2021.

Bak, P., Tang, C., and Wiesenfeld, K.: Self-organized criticality: An explanation of the 1/f noise, Phys. Rev. Lett., 59, 381, 1987.

Buaria, D., Pumir, A., Bodenschatz, E., and Yeung, P.: Extreme velocity gradients in turbulent flows, New J. Phys., 21, 043 004, 2019.

180   Dubrulle, B.: Intermittency in Fully Developed Turbulence: Log-Poisson Statistics and Generalized Scale Covariance, Phys. Rev. Lett., 73, 959–963, 1994.

Horbury, T. and Balogh, A.: Structure function measurements of the intermittent MHD turbulent cascade, Nonlinear Proc. Geoph., 4, 185–199, 1997.

Lovejoy, S., Schertzer, D., and Stanway, J. D.: Direct Evidence of Multifractal Atmospheric Cascades from Planetary Scales down to 1 km,

185   Phys. Rev. Lett., 86, 5200–5203, 2001.

Politano, H. and Pouquet, A.: Model of intermittency in magnetohydrodynamic turbulence, Phys. Rev. E, 52, 636–641, 1995.

Sardeshmukh, P. D. and Sura, P.: Reconciling Non-Gaussian Climate Statistics with Linear Dynamics, J. Climate, 22, 1193–1207, 2009.

Schekochihin, A. A.: MHD Turbulence: A Biased Review, J. Plasma Phys., 88, 155880 501, 2022.

Yeung, P. K., Zhai, X. M., and Sreenivasan, K. R.: Extreme events in computational turbulence, PNAS, 112, 12 633–12 638, 2015.

190   Yeung, P. K., Sreenivasan, K. R., and Pope, S. B.: Effects of finite spatial and temporal resolution in direct numerical simulations of incompressible isotropic turbulence, Phys. Rev. Fluids, 3, 064 603, 2018.

---

## Author Response (AR3)

**Intermittency in fluid and MHD turbulence analyzed through the prism of moment scaling predictions of multifractal models**

A. Pouquet et al.

April 16, 2025

   **\*\*\***     This is response 2 to Shaun Lovejoy.

All remarks have been taken into account, with changes in this second revision marked in blue, see p. 1, 5, 14–16.

   **\*\***   Figure 4 has been remade with larger fonts and better balance in [X,Y].

   **\*\***   The monofractal limit of $a_Q$ was already mentioned in the text (see line 263), and it is indeed 3, as also found previously in Sardeshmukh et al. (2009). The data in Fig. 4 is not in great agreement with this, but not too bad either.

   **\*\***   Concerning the more general comment, and in particular that of referee 2 number 10, we have added a text (see conclusion), and some references in so doing. We mention that indeed anomalous scaling exponents of structure functions in time should be computed, and their $\infty$ limit evaluated; this is left for future work. As for the spectrum of singularities (with even unboundedness possibly linked to lognormality, posing the question of lognormal or log-Lévy processes), that would require substantially more work and it will not be undertaken now, for one thing the paper being quite long already.